# PATCHBLENDER: A MOTION PRIOR FOR VIDEO TRANSFORMERS

## ABSTRACT

Transformers have become one of the dominant architectures in the field of computer vision. However, there are yet several challenges when applying such architectures to video data. Most notably, these models struggle to model the temporal patterns of video data effectively. Directly targeting this issue, we introduce Patch-Blender, a learnable blending function that operates over patch embeddings across the temporal dimension of the latent space. We show that our method is successful at enabling vision transformers to encode the temporal component of video data. On Something-Something v2 and MOVi-A, we show that our method improves the baseline performance of video Transformers. PatchBlender has the advantage of being compatible with almost any Transformer architecture and since it is learnable, the model can adaptively turn on or off the prior. It is also extremely lightweight compute-wise, 0.005% the GFLOPs of a ViT-B.

## 1 INTRODUCTION

The Transformer (Vaswani et al., 2017) has become one of the dominant architectures of many fields in machine learning (Brown et al., 2020; Devlin et al., 2019; Dosovitskiy et al., 2020). Initially proposed for natural language processing (Vaswani et al., 2017), it has since been shown to outperform convolutional neural networks in the image domain (Dosovitskiy et al., 2020). Adapting such vision models to the video domain has been straightforward and resulted in new state-of-the-art results (Arnab et al., 2021). Since then, multiple Transformer based methods have been proposed (Bertasius et al., 2021; Fan et al., 2021; Liu et al., 2021), making steady progress on a variety of challenges in the video domain.

Despite these advances, there are still many challenges when it comes to applying Transformers to video data. One such challenge is the lack of a strong inductive bias in the Transformer architecture with respect to temporal patterns. This challenge is most evident in the attention mechanism, where it can be difficult for patches to attend to relevant patches across time. Picture for example multiple frames of a blue sky with birds, how is a given patch supposed to attend to its relevant spatial location in each frame? If it can't do so, how would it know if a bird was present in the patch at some point in time? This issue makes it difficult for Transformers to properly model video data.

To address the challenge of modeling temporal information in video data, we propose a new temporal prior for video transformer architectures. This novel prior, called PatchBlender , is a learnable smoothing layer introduced in the Transformer. The layer allows the model to blend tokens in the latent space of video frames along the temporal dimension. We show that this simple technique provides a strong inductive bias with respect to video data, as it allows for easier mapping of relevant spatio-temporal patches in the attention mechanism.

We evaluate our method on three video benchmarks. Experiments on MOVi-A (Greff et al., 2022) show that Vision Transformers (ViT) (Dosovitskiy et al., 2020) with PatchBlender are more accurate at predicting the position and velocity of falling objects compared to the baseline. For Something-Something v2 (Goyal et al., 2017), we also find that PatchBlender improves the baseline performance of ViT and MViTv2 (Li et al., 2021). We also include results on Kinetics400 and show that PatchBlender learns to weakly exploit the temporal aspect of Kinetics400. Specifically, unlike Something-Something v2 and MOVi-A, it has been shown that one can achieve competitive performances on this dataset even without temporal information (Fan et al., 2021; Sevilla-Lara et al., 2019). Interestingly, we provide further evidence to this, as we show that our PatchBlender

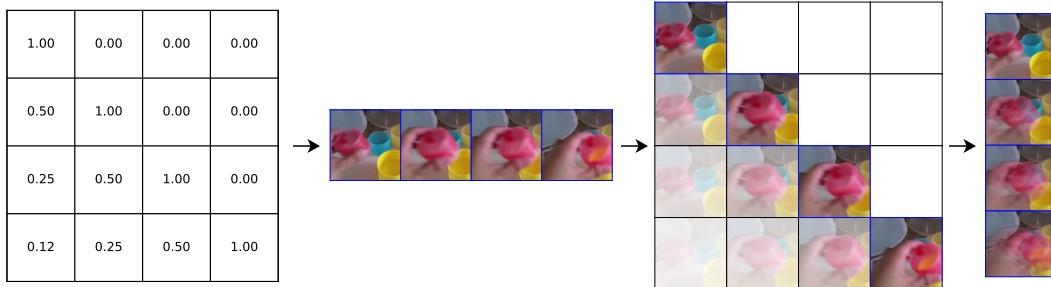

Figure 1: Illustrative example of PatchBlender. On the left, we have an example of learned blending ratios, where the diagonal represents the frame being blended and the other values in a given row correspond to the blending ratios of the other frames. We then apply these ratios to a sequence of four frames which gives us a pattern blending each frame with their past frames. The final result is a sequence of four frames, where each frame is lightly blended with past frames, depicting the motion leading to that point in time. Note that while this example is with RGB frames, our method is actually applied to the latent representation of these frames within the Transformer.

learns an identity-like function as the optimal smoothing operation. Finally, our method is also very lightweight compute-wise, adding only 0.005% to the total GFLOPs of a ViT-B.

## 2 RELATED WORK

Several prior work have explored how to best handle temporal data in machine learning, e.g., using the convolutional neural networks Ji et al. (2013). Notably, Feichtenhofer et al. (2018) propose to have a combination of two networks, one operating at a low frame rate to model long term dependencies, while the other operates at a fast frame rate to model local dependencies. Liu et al. (2020) also propose adaptive temporal kernels to better model complex temporal dynamics.

With respect to Transformers, one popular approach for incorporating a spatial-temporal bias to the model has been to process the data in a hierarchical manner Arnab et al. (2021); Chen et al. (2021); Fan et al. (2021); Li et al. (2022); Liu et al. (2021); Yan et al. (2022); Zha et al. (2021). Other work have explored modifying the attention mechanism in order to enforce a temporal bias Bertasius et al. (2021); Bulat et al. (2021); Guo et al. (2021); He et al. (2020); Patrick et al. (2021); Zhang et al. (2021).

Another type of approach has been to incorporate motion information into the model in an explicit form . Chen & Ho (2021) process all the information available from raw video data, which includes motion and audio. Wang & Torresani (2022) propose to use motion information in order to determine where to attend to in the Transformer's attention mechanism. Other non-Transformer work which make use of motion typically consist of a two stream network, with the RGB and motion data handled seperately Diba et al. (2016); Feichtenhofer et al. (2016b); Girdhar et al. (2017); Gkioxari & Malik (2014); Simonyan & Zisserman (2014) or jointly throughout the network Feichtenhofer et al. (2016a); Feichtenhofer et al. (2017); Jiang et al. (2019); Wang et al. (2019); Zhang et al. (2016).

In comparison with all these methods, our work is novel just by definition. We do not change the Transformer's architecture into a hierarchical one or change the attention mechanism itself. Instead, we introduce a new layer which can be inserted almost anywhere in the Transformer. It takes the latent representation of the frames at the step where the layer is inserted and blends them along the temporal dimension. The layer is thus compatible with almost any Transformer variant and attention mechanism. As long as there is a latent representation for each frame or each patch, they can be blended.

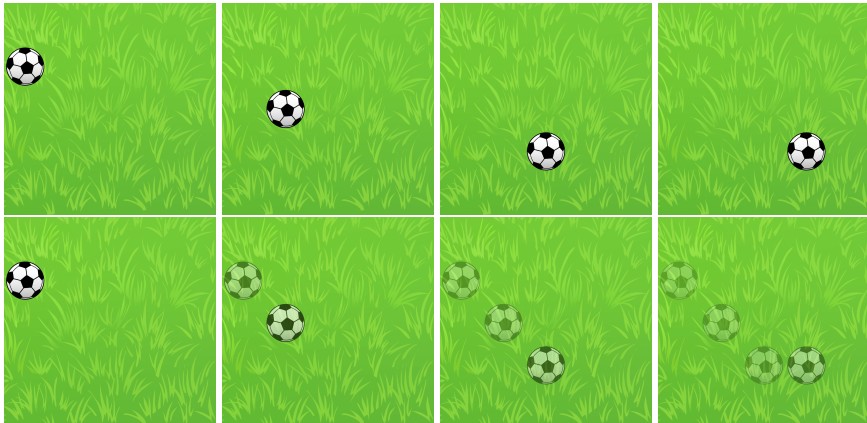

Figure 2: Top: four frames representing a ball moving across a grass field. Bottom: the same four frames, temporally smoothed. The latter provides a stronger signal with respect to the temporal nature of the data and can help the Transformer attend to relevant spatial-temporal locations in each frame.

## 3 METHODOLOGY

We begin by explaining why the lack of a temporal prior is a key challenge with Transformers when modelling video data. We then describe our PatchBlender, which is a simple way to incorporate motion priors to video transformers as a temporal inductive bias for video data.

### 3.1 MOTIVATION: TRANSFORMERS AND TEMPORAL DATA

An issue faced by Transformers when applied to video data is the architecture's lack of an effective temporal inductive bias. The only mechanism in a Transformer which allows it to deduce the sequence order is the positional embedding which is summed with the input sequence. This signal is typically weakly exploited by Transformer models, as performance is usually unimpaired when the positional embedding is removed (Bertasius et al., 2021) or if frames are shuffled (Fan et al., 2021; Sevilla-Lara et al., 2019).

We can visualize how a stronger inductive bias can potentially make it much easier to model the motion of scene elements. In Figure 2, in the top row, we have four frames of a ball moving across a grass field. If we were to smooth the frames, as in the bottom row of Figure 2, the signal with respect to motion would potentially be stronger. For example, simply looking at how faded an object is in each patch can indicate how long this object was at those spatial locations and potentially make it easier to track its motion.

With respect to Transformers, this can also potentially affect how patches attend each other. Specifically, in our example, the ball in each frame can potentially attend more easily to its previous spatial location in the same frame. Moreover, since patches have been blended with the patches in the same location in each frame, they can also potentially attend more easily to that spatial location in each frame.

### 3.2 PATCHBLENDER

We propose to add a temporal inductive bias to video transformers to better model video data. To this end, we develop PatchBlender, a learned temporal smoothing function for the Transformer's latent space. The goal is to allow a given patch to attend more easily to the relevant spatial location in other frames, as well as to provide the model with a better sense of how scene elements are moving across time. To achieve this, we introduce a new layer which blends the latent representation of the frames, more specifically their patches, along the temporal axis.

The PatchBlender layer takes as input the latent representation $X$, a tensor of shape $n \times p \times z$, where $n$ is the number of frames, $p$ is the number of patches per frame and $z$ is the embedding size of each

Table 1: PatchBlender improves the performance of a ViT-B and MViTv2-S on Something-Something v2 and of a ViT-B on MOVi-A. Performance for Kinetics400 and Something-Something v2 is top-1 accuracy, while MOVi-A is mean squared error loss.

| Method | MOVi-A Pos. ↓ | MOVi-A Vel. ↓ | SSv2 ↑ | Kinetics400 ↑ |
|---|---|---|---|---|
| ViT-B | 0.42 | 1.40 | 62.26 | **77.82** |
| ViT-B + PatchBlender | **0.22** | **0.96** | **62.53** | 77.62 |
| MViTv2-S | - | - | 66.34 | - |
| MViTv2-S + PatchBlender | - | - | **66.49** | - |

patch. It then applies the blending function $b$:

$$b(X) = RX \tag{1}$$

where $R$ are the learnable blending ratios, a matrix of size $n \times n$ and the output of $b(X)$ is a tensor $\tilde{X}$ of the same shape as $X$. The $i$th output frame in $\tilde{X}$ is thus the result of multiplying each frame in $X$ with the corresponding ratios in row $R_i$ and summing the result:

$$\tilde{X}_i = \sum_{j=1}^{n} R_{i,j} X_j \tag{2}$$

Hence, if $R$ would be an identity matrix, $b(X)$ would simply be the identity function.

The blending ratios in $R$ are learned through training, enabling the model to adaptively decide on the details of smoothing depending on input patterns. The model could learn for example, to blend in both directions or to only blend backwards or forwards. It can also control the range of the blending operation and like previously mentioned it could learn to completely turn the prior off by learning the identity function. No restrictions are enforced on the possible values nor are they normalized prior to blending.

Our layer can be inserted at any step in the Transformer, as long as its input respects the specified shape. The layer could also be applied to the initial RGB input, but information would be lost through the process. In comparison, having PatchBlender in a Transformer layer, the input of the Transformer layer is preserved in the skip connection. This allows us to blend the latent representation and then summing back the input into the latent representation, after the Transformer layer. We can thus for example blend before the attention mechanism, perform attention, feedforward and then add back the unblended latent representation.

Different PatchBlender layers could also be inserted at multiple steps in a Transformer. This could enable the model to learn different blending functions at each layer and how to combine them. If blending at a certain layer would be detrimental, then the model could simply learn to turn off the prior for that layer.

## 4 EXPERIMENTS

We begin by evaluating the effect of our PatchBlender on the performance of video Transformers, namely, Vision Transformers (ViT) (Dosovitskiy et al., 2020) and Multiscale Vision Transformers (MViT) (Li et al., 2021). In this regard, we train ViT-B models with PatchBlender on three video datasets: MOVi-A (Greff et al., 2022), Something-Something v2 (Goyal et al., 2017) and Kinetics400 (Kay et al., 2017) and compare the performance with a baseline ViT-B. Similarly, we ran MViTv2-S experiments with PatchBlender specifically on Something-Something v2. The ViT-B architecture consists of 12 Transformer layers. We add a PatchBlender layer to its second Transformer layer, right before the self-attention. As for the MViTv2-S, it consists of 16 Transformer layers. We add a PatchBlender layer to all of them, also right before the self-attention. We empirically found this to perform best. Blending ratios are initialized with the identity matrix $I$.

Table 1 summarizes our results. Details and analysis are provided for each benchmark in the following sections.

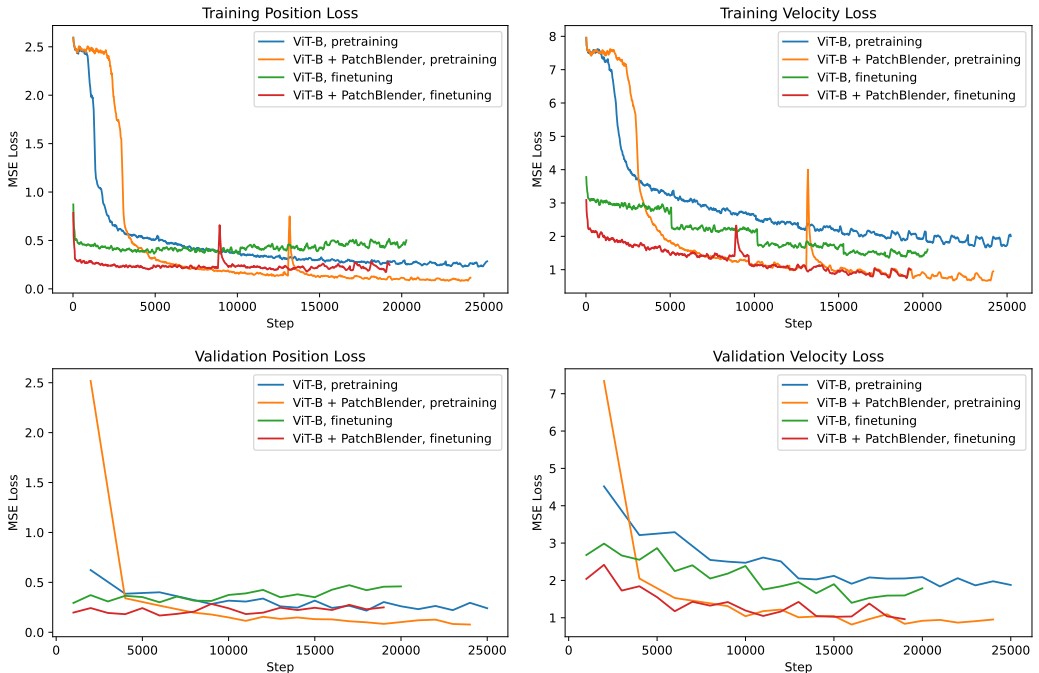

Figure 3: MOVi-A loss comparison. In the pretraining task, the model must predict, for a randomly selected object, its position and velocity in the shown frames. For finetuning, the model must predict, for a randomly selected object, its position and velocity in future frames. Our method improves the performance for both the pretraining and finetuning task.

## 4.1 MOTION PREDICTION ON MOVI-A

The MOVi-A dataset[1] (Greff et al., 2022) consists of 9703 train and 250 validation videos of objects falling in a 3D environment. The videos are two seconds long, at a frame rate of 12 FPS. We chose this dataset to show that our method can help in a setting different than the standard action recognition benchmarks and evaluate on a task which focuses purely on modeling movement through time. The dataset provides randomly generated scenes of random objects with random trajectories. For each frame, annotations provide the position and velocity of each object in the scene, as well as other information. We wanted the model to learn to predict future movement based on the past. Therefore, we design the following task formulation: given $n$ number of frames, predict the position and velocity of a randomly picked object in the scene at frame $n + t$. In practice, we set $n$ to 8 and $t$ to 4, which corresponds to $\frac{1}{3}$ of a second after the end of the shown clip.

**Training and evaluation.** Our baseline model is the ViT-B variant of the Vision Transformer (Dosovitskiy et al., 2020). We use ViT-B models pretrained on ImageNet21k (Russakovsky et al., 2014). To get the model to predict future position and velocity of an object in the scene, we had to first train it to predict these values for the given frames. In order to do this, we draw a bounding box around the randomly picked object and make the classification head output $8 \times 6$ values, corresponding to the 3D world coordinates and velocity of the bounded box object in each frame. We then apply mean squared error loss against the target values. Frames are sampled by first randomly selecting a frame between the first and the fourth included, then sampling the seven frames following the randomly selected frame. At inference, prediction is done using a single clip, center cropped. The classification head has a hidden size of 768, with no dropout. Adam is used with a base learning rate of 0.001. The learning rate is not tuned with a schedule. Batch size is 128. Weight decay is set to zero.

---

[1]https://github.com/google-research/kubric

Once the model has been trained to predict the position and velocity of the bounded box object in the shown frames, we begin the second-phase of training, where now the model must predict these values for the frame following the last shown frame. The classification head now only outputs 6 values, as the loss corresponds to the future position and velocity. As the model trains for this task, we iteratively increase the "distance" of the frame to predict by one, until the model must predict the values of the fourth frame following the last shown frame.

**Results.**    Training and validation performance are shown in Figure 3. We denote the "pretraining" step as the prediction of the position and velocity for the shown frames and "finetuning" as the prediction of the position and velocity for future frames. Loss is better in both cases for the ViT-B with PatchBlender.

We can see the effect of increasing the "distance" of the frame to predict in the training velocity loss, where the one for the ViT-B model resembles a step function diminishing each time the distance is increased and then remains relatively flat. Interestingly this same effect is not observed for the ViT-B with PatchBlender, where the loss instead seems to go down more smoothly.

When increasing the frame distance, we also observe that the position loss seems to increase compared to the diminishing velocity loss. This could be explained by the velocity of the objects slowing down through the video, making it easier to predict, and/or potentially objects leaving the camera view, making it harder to predict their position.

For the final step of the finetuning, predicting the position and velocity of the fourth frame after the last shown frame, we report the best validation loss in Table 1.

## 4.2    ACTION RECOGNITION ON SOMETHING-SOMETHING V2

The Something-Something v2 (SSv2) dataset [2] (Goyal et al., 2017) is an action-recognition benchmark consisting of 168,913 train and 24,777 validation videos of human-object interactions, with a camera centered on the action taking place. The dataset also contains 27,157 unlabeled test samples, which we make no use of. Videos are typically a few seconds long, at a frame rate of 12 FPS. Labels are more fine-grained and less diverse than in Kinetics400 Kay et al. (2017) elevating the importance of the temporal aspect. Some example labels: "Pushing something from left to right," "Pushing something from right to left," "Covering something with something," "Uncovering something." Based on the nature of the data, we believe this dataset to be a good benchmark for evaluating a model's ability to model temporal information.

**Training and evaluation.**    Our training and evaluation procedure for SSv2 is the same as with Kinetics400, see section 4.3, except for the following differences. Clips are sampled by first splitting the video into 8 equally sized bins and then sampling one frame from each bin. Random horizontal flipping of frames is applied when this does not corrupt the label. For example, videos with the label "Pulling something from left to right" or "Pulling something from right to left" cannot be horizontally flipped, while videos with the label "Throwing something" can. At inference, prediction is averaged over 5 different spatial crops. The classification head's hidden layer size is 768 instead of 3072. We train the model for 33k steps with a batch size of 512 and a base learning rate of 0.02. We initialize the models with weights pretrained on Kinetics400.

**Results.**    Prediction accuracy on the validation set is reported in Table 1. We find that PatchBlender improves the top 1 accuracy of the ViT-B by 0.27% and of the MViTv2-S by 0.15%. This is a slight improvement, but still significant given that improvements for state-of-the-art methods on Something-Something v2 tend to be marginal (Wang & Torresani, 2022). PatchBlender might be useful for this dataset, as it contains many labels for which understanding the temporal order is important. These results as well as a qualitative analysis in section 4.4, suggest that our learnable prior helps Transformers better model such type of problems.

---

[2]`https://developer.qualcomm.com/software/ai-datasets/`
`something-something`

### 4.3 ACTION RECOGNITION ON KINETICS400

The Kinetics400 (Kay et al., 2017) dataset consists of around 240k train and 20k validation clips of about 10 seconds, taken from YouTube videos. The frame rate is typically 30 FPS. Labels cover a broad range of activities, such as "eating ice cream," "ski jumping," and "planting trees." We include results on this benchmark, as is standard practice in the field of vision, but provide further evidence in section 4.4 to prior evidence (Sevilla-Lara et al., 2019; Neimark et al., 2021; Fan et al., 2021) that temporal information is unimportant for this benchmark.

**Version inconsistencies.**   As there is no official copy available online, the Kinetics400 videos must be downloaded through YouTube in order to obtain the dataset. This is an issue since some of these videos have become unavailable through time. Consequently, this has lead to inconsistencies in the versions used by various research groups, in both number of samples, as well as clip length. Our own copy consists of 240,708 train and 19,795 validation clips. Fortunately, we observe a ViT-B baseline performance which closely matches what other research groups have previously reported (Fan et al., 2021). For a fair comparison though, we compare PatchBlender results with our own baseline trained on the same version of Kinetics400.

**Training and evaluation.**   Our baseline model is the ViT-B variant of the Vision Transformer (Dosovitskiy et al., 2020). For both the baseline and the ViT-B with PatchBlender , we train the models using the following hyperparameters. We train the models with clips of 8 frames, sampled by randomly selecting the first frame and then applying a stride of 8 to select the remaining 7 frames of the clip. During training, the same random crop of 224 $\times$ 224 is applied to all frames of the clip, as well as a random horizontal flip. At inference, prediction is averaged over 5 different, center cropped clips of the same video. Frames are split into patches of 16 $\times$ 16 pixels. The classification head of the model has a hidden layer of size 3072, a Tanh activation and dropout (Srivastava et al., 2014) of 0.5. The models are pretrained on ImageNet21k and trained for 28k steps, with a batch size of 256. SGD is used, with a base learning rate of 0.01, a cosine annealing schedule, momentum of 0.9 and weight-decay is set to 0.0001.

**Results.**   Performance on the validation set is reported in Table 1. We observe no benefit to using PatchBlender for Kinetics400. This result is unsurprising to us. Indeed, as previously mentioned, Neimark et al. (2021) and Fan et al. (2021) have reported that shuffling the frame order, which completely destroys the temporal information of the video, does not impair the prediction accuracy on Kinetics400. The reason for this, we believe, is simply that the dataset's label space is constructed in a way that makes temporal information matter less. Indeed, prior work have demonstrated strong scene bias issues in this dataset Sevilla-Lara et al. (2019). Looking at only one frame, one can easily guess the correct label. This does not serve us well if our goal is to evaluate the ability to model information through time. While Kinetics400 is a popular video benchmark, we believe it to be a poor choice when evaluating motion priors. We provide further evidence to this claim in the following section.

### 4.4 QUALITATIVE ANALYSIS

We examine the blending functions learned by the ViT models for all three benchmarks evaluated on. The blending ratios $R$ are shown in Figure 4.

**MOVi-A.**   For MOVi-A, the learned blending function is the left heatmap in Figure 4. This is the learned blending function of the finetuned model. In comparison with Something-Something v2, PatchBlender learned to positively blend mostly future frames and negatively past frames. Interestingly, this happens for the first four frames. As for the last four, they seem to negatively weigh both prior and future frames. It is possible that since both velocity and position must be predicted, the first four frames are used to better predict the velocity, while the last four frames might be useful to predict the final position of the object. Like previously mentioned, objects in MOVi-A scenes do sometimes leave the camera view sooner than the end of the clip.

**SSv2.**   For Something-Something v2, the learned blending function is the center heatmap in Figure 4. In the case of frame 0, we can see that almost all of the weight is given to itself and all future

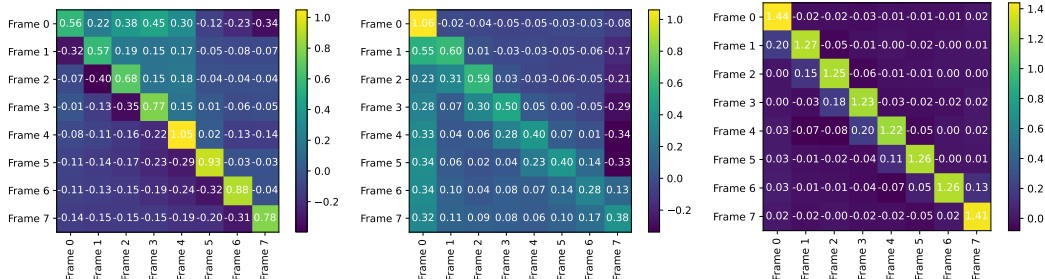

Figure 4: Visualization of the learned PatchBlender ratios per dataset, (left) MOVi-A, (center) Something-Something v2, (right) Kinetics400. We can see that for both MOVi-A and Something-Something v2, the model learns to exploit the temporal nature of the data by blending frames with past and future frames. In contrast, for Kinetics400, the temporal aspect of the data seems to be mostly unimportant to the model.

Table 2: Shuffling frames does not impair the performance of a vision Transformer on Something-Something v2, unlike with our temporal prior, which shows that with our method, the model learns to exploit the temporal nature of the data.

| Method | Top 1 | |
| | No Shuffling | Frame Shuffling |
| --- | --- | --- |
| ViT | 41.57 | 42.20 |
| ViT + PatchBlender | 48.69 | 43.02 |

frames are weighted negatively. Interestingly, the model chose to apply positive weights mostly to prior frames and ignore or negatively penalize future frames. This indicates the importance of the temporal dimension of the data to the model. Another interesting aspect is that the first frame is always highly weighted in comparison to other frames, sometimes even more than the frame being smoothed. We believe that this is because the model is using the first frame as a reference. There are many opposite labels in Something-Something V2, such as 'Moving something up' and 'Moving something down', and so knowing where something is in relation to where it started is important. We also see that the frames close to the frame being smoothed have more importance than prior ones. This could be because it helps the model to understand the motion of objects in the scene at each given frame.

**Kinetics400.** The PatchBlender layer trained on Kinetics400 is the right heatmap in Figure 4. As we can see, the model almost learns a function very close to the identity function and barely exploits any temporal information. The only noteworthy blending happens for frames 2 to 7, where from 2 to 6 the information from the previous frame is slightly used. For frame 7, the same can be said but using the following frame. For us, this highlights how unimportant temporal information is for this dataset, in comparison to the functions learned for Something-Something v2 and MOVi-A.

Overall, it is interesting to see that for all three benchmarks, PatchBlender learned blending functions unique to each. The temporal information is exploited differently in all three cases, highlighting the adaptability and usefulness of our learnable temporal prior.

## 4.5 ABLATION STUDY

In this section, we perform multiple ablation experiments in order to better understand how our PatchBlender helps Transformer to model temporal information. Due to the high computational cost involved by these ablations, we limit these evaluations to a smaller variant of the ViT architecture on the SSv2 benchmark.

**Temporal Information Importance.** As a first test, we verify the importance of temporal information in models trained with PatchBlender. Therefore, we randomly shuffle the input frames and

Table 3: PatchBlender variations on SSv2. The best performance is achieved when patches are blended with patches in the same location in each frame. This highlights our original motivation that helping the model to better understand the spatial-temporal aspect of the data is important.

| Method | Top 1 |
|---|---|
| ViT | 41.57 |
| ViT + PatchBlender, blending with random patches in each frame | 45.40 |
| ViT + PatchBlender, blending with the same random patch location in each frame | 44.49 |
| ViT + PatchBlender | 46.55 |

look at performance degradation of the ViT model with and without PatchBlender. Table 2 shows that this results in a drop of 5.67 points. In contrast, the baseline ViT is unimpaired by the frame shuffling, indicating that the model does not make use of temporal information. This highly suggests that adding PatchBlender to the model results in it using temporal information.

**Blending Variations.** The next series of tests measures whether blending along the same patch location is important or if simply spreading the latent information throughout the sequence is enough to improve the performance. In order to evaluate this, we evaluate the following variants. Our first variant blends patches not at the same location in each frame, randomly picking the patch location in each frame. Our second variation is similar, but picks one random patch location instead of different ones in each frame. We compare both these variants to a baseline ViT and a ViT with the standard PatchBlender. Results are shown in Table 3.

We can see that blending along the same patch location performed best. Interestingly, both other variants of blending performed better than the baseline ViT, indicating that spreading information temporally throughout the latent representation does help the model. Blending with random patches in each frame performed better than choosing the same patch location at random. This could be because the motion prior has a higher chance of containing patch information from a spatially close patch to the one being smoothed compared to the latter case. It could also be because the prior blends with a more diverse set of patches, which could be useful for the model.

## 5 CONCLUSION

We proposed a new motion prior for the Transformer, with the goal of better modeling video data. Our results show that PatchBlender, a simple and lightweight add-on, is useful in that respect.

PatchBlender improves the performance of the Vision Transformer and Multiscale Vision Transformer on both MOVi-A and Something-Something v2 and Something-Something v2 respectively. However, we observe a slight degradation of performance on Kinetics. Further study is required to understand why this is the case.

In other respects, it is uncertain if PatchBlender can be useful when there is significant camera move- ment. As a future work, we are interested in evaluating how well PatchBlender generalizes to various types of video Transformers and also with datasets with notable camera movement.

Overall, we believe that designing better motion priors in order to model video data is a challenging problem. We hope that our work will spark some useful ideas in the community in that respect.

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
