# OpenReview forum: "PatchBlender: A Motion Prior for Video Transformers"
_ICLR.cc/2023/Conference — Submitted to ICLR 2023_

### Official Review · Reviewer_4JAh · 2022-10-19

**Confidence:** 4
**Clarity, Quality, Novelty And Reproducibility:** 1 Clarity
**Correctness:** 3
**Technical Novelty And Significance:** 2
**Empirical Novelty And Significance:** 2
**Recommendation:** 3

**Strength And Weaknesses:**

1 Strength
1) The paper is well written.
2) The temporal modeling in ViT is an important problem for video representation learning.

2 Weakness
1) The novelty is relatively limited. Such PatchBlender is quite similar to temporal convolution. Hence, the design is bascially to insert a temporal-conv-like layer into the spatial ViT. Such architectures are not new in the literature such as UniFormer.
2) The experiment is weak to support the claim.
2.1) It would be interesting to show the experiment where temporal convloution is inserted into ViT.  This is an opportunity to show what is the difference between two operations, and which one is better.
2.2) It lacks the state-of-the-art comparison on these video benchmarks.

**Summary Of The Paper:**

In this work, the authors propose to tackle temporal modeling in video data. To achieve this goal, they introduce a PatchBlender layer, which uses a learnable matrix to describe the temporal dynamics among frames. They use MOVi-A, Something-Something v2 and Kinetics for model evaluation.

**Summary Of The Review:**

The paper considers an important problem in video understanding. However, the proposed operation is similar to temporal convolution, and the experiments ate weak to support the claim.

---

> ### Author Response · Authors · 2022-11-16
> **Clarifications to some of the comments made by the reviewer**
>
> i. "It would be interesting to show the experiment where temporal convloution is inserted into ViT. This is an opportunity to show what is the difference between two operations, and which one is better."
>
> We agree that exploring Uniformer style convolutions but in the ViT could be an interesting comparison. We would like to clarify though that although ours and theirs are temporal convolutions, they are not the same operation. The Uniformer kernels look at 5 frames at most, whereas in our case, the kernel looks at all the frames in the sequence. Additionally, the kernel size in the Uniformer are also different with respect to the height and width. Theirs is much smaller, at most 5x5, whereas our kernel size covers the entire patch, which is 16x16.

---

### Official Review · Reviewer_4WoX · 2022-10-23

**Confidence:** 5
**Correctness:** 3
**Technical Novelty And Significance:** 2
**Empirical Novelty And Significance:** 2
**Recommendation:** 3

**Clarity, Quality, Novelty And Reproducibility:**

This paper is clearly organized and the experiments are easy to reproduce. Considering the similarity of PatchBlender and the temporal linear layer, the novelty of the work is limited. Most of the experiments are conducted on simple MOVi-A, and the quality of experiments is doubted.

**Strength And Weaknesses:**

Strength:
1. The design of PatchBlender is simple yet effective for MOVi-A.
2. The visualizations clearly show the motivation of PatchBlender.
3. The experiment details are provided and easy to follow.

Weakness
1. Considering MOVi-A is an easy benchmark without complicated object interaction and camera movement, the real improvement of PatchBlender for daily activity recognition is doubted. As expected, the improvement on Something-Something v2 is marginal. And it even decreases the performance on Kinetics.
2. The paper only conducts experiments based on ViT, thus the generality is not well-demonstrated. For example, does PatchBlender also help popular video transformer like TimeSformer[1], VideoSwin[2], and UniFormer[3]? In my opinion, those models with temporal operation may not be complementary to PatchBlender.
3. The PatchBlender is actually a learnable linear layer conducted in the temporal dimension. Considering the optimization problem, it may not work as well as simpler temporal convolution, which has been demonstrated to be effective in previous 3D CNNs.

> [1] Gedas Bertasius, Heng Wang, and Lorenzo Torresani. Is space-time attention all you need for video understanding? In International Conference on Machine Learning, 2021.
>
> [2] Ze Liu, Jia Ning, Yue Cao, Yixuan Wei, Zheng Zhang, Stephen Lin, and Han Hu. Video swin transformer. In IEEE/CVF Conference on Computer Vision and Pattern Recognition, 2022.
>
> [3] Kunchang Li, Yali Wang, Gao Peng, Guanglu Song, Yu Liu, Hongsheng Li, and Yu Qiao. Uniformer: Unified transformer for efficient spatial-temporal representation learning. In International Conference on Learning Representations, 2022.

**Summary Of The Paper:**

To prompt token interaction over time, this paper proposes PatchBlender, a learnable matrix to mix tokens over time. Via inserting the PatchBlender in the middle layers of ViT, it improves the ability for temporal modeling. Experiments on MOVi-A and Something-Something v2 support the effectiveness of PatchBlender.

**Summary Of The Review:**

The paper proposes a simple PatchBlender, which is plug-and-play and helpful for temporal modeling. However, it seems PatchBlender only works for vanilla ViT and the easy benchmark without complicated object interaction and camera movement. Considering the diversity of human action, the effectiveness of PatchBlender for daily activity is limited, which is also represented in the unsatisfactory results on Something-Something and Kinetics. Besides, it may not work for the popular video transformer due to the simple design of the temporal linear layer. More experiments of different backbones and more comparisons are needed to demonstrate its generality and effectiveness.

---

> ### Author Response · Authors · 2022-11-16
> **Clarifications to some of the comments made by the reviewer**
>
> i. "Considering MOVi-A is an easy benchmark without complicated object interaction and camera movement, the real improvement of PatchBlender for daily activity recognition is doubted. As expected, the improvement on Something-Something v2 is marginal. And it even decreases the performance on Kinetics."
>
> We believe that this is very subjective. Yes MOVi-A has no camera movement, but we would not claim that SSv2 or Kinetics400 have more complicated object interaction or that they are more "difficult" benchmarks. All these benchmarks are useful in their own way and each have their own respective properties. As discussed in the paper, we believe that Kinetics400 is not very interesting from a "modelling motion" perspective, as you can get very good accuracy on this video benchmark using only a single frame and the performance for most classes is also unaffected by shuffling the frame order. While we believe that SSv2 is a better choice than Kinetics400 to evaluate the ability to model motion, it is an action recognition dataset and it is thus possible that models can get good performance on this dataset while not necessarily learning how to model motion through time. This is why we decided to use MOVi-A. The task is specifically to predict motion, so it is a direct evaluation of the property we cared about.
>
> ii. "it may not work as well as simpler temporal convolution"
>
> We are aware of temporal convolutions being used in Transformers. However, we do not agree with the reviewer's point that there are temporal convolutions which are "simpler" than our method, given that PatchBlender is a linear operation and can be defined as a standard convolution. Our method is learnable though. If the reviewer is referring to fixed linear operations, then yes, it is possible that those could work better than a learned linear operation. From our limited experiments with fixed blending ratios, we did not find anything which worked better than learned ratios. We did not compare though with other fixed linear operations that the reviewer may be referring to.

---

### Official Review · Reviewer_rJNz · 2022-10-28

**Confidence:** 4
**Correctness:** 3
**Technical Novelty And Significance:** 2
**Empirical Novelty And Significance:** 2
**Recommendation:** 5

**Clarity, Quality, Novelty And Reproducibility:**


The paper is well presented. The authors performed thorough experiments, attempting to show the pros of the proposed approach. The results are well illustrated and analysed. The approach is simple  and well described, it could be re-implemented by a graduate students.


**Details Of Ethics Concerns:**

No ethic concern

**Strength And Weaknesses:**


Learning a motion prior is certainly a long-standing open issue. The approach proposed by the author is extremely simple (learn a weight matrix). Several applications are illustrated and experimental results developed.

The first set of experiments (Movi-A) seems controversial to me. Given a single frame as input, the prediction of future motion of a given object is not unique (there is no deterministic motion from a single frame, there are potentially an infinity of plausible motions), at least in real videos. Hence, I do believe that the prior learnt in this context is not what is expected in real life and that quantitative results in this set of experiments are irrelevant. The learnt motion prior will however certainly 'remenber' simple dynamics, a similar task that in SSv2.

In SSv2, the quantitative improvement between ViT and ViT+PatchBlender is not significant (0.27%). It would have been interesting to plug-in the PatchBlender onto several other approaches dedicated to video labeling (most are now using Transformers in a way or in an other) and to report these additional results.

It seems to me that although the idea of linear combination of temporal embedings via the learnt weights matrix R might have some interest, it might be a very 'rigid' prior, ie limited to learn very generic dynamics (eg, go right, left, etc), but unable to cope with more complex motions of semi-rigid objects (ie human activity) or of multiple objects. It would be interesting to discuss the limitations of the approach.


**Summary Of The Paper:**


The authors aim at learning a generic motion prior, to ease different video-related tasks (motion prediction, video labeling). They argue that learning the linear combination of temporal embeddings of a Transformer from videos enables to encode the dynamic of an observed moving object. The approach is validated on motion prediction from a single input frame (synthetic dataset) and video labelling (on two classic benchmarks). Results show the effect of such a prior.

**Summary Of The Review:**

Learning motion prior is a super interesting subject. The merit of the present work is to try to validate the approach on different tasks, to illustrate the generability of the prior. The proposed weight matrix however does not seem  to do a significantly better job that the transformer itself.

---

> ### Author Response · Authors · 2022-11-16
> **Clarifications to some of the comments made by the reviewer**
>
> i. "The first set of experiments (Movi-A) seems controversial to me. Given a single frame as input, the prediction of future motion of a given object is not unique"
>
> We would like to clarify that for the MOVi-A task, as stated in the paper, the model must predict future motion using n previous frames, not just a single frame. We typically give the first 8 frames to the model and ask to predict the motion of the 12th frame.
>
> ii. "it might be a very 'rigid' prior, ie limited to learn very generic dynamics (eg, go right, left, etc), but unable to cope with more complex motions of semi-rigid objects (ie human activity) or of multiple objects."
>
> We do not believe this to be the case based on how our method works. If we take one patch as an example, in frame t, our method can learn to take information from the same patch in previous and future frames and merge it with the current patch. This helps modelling what was in that particular spatial location across time. The complexity of the motion is irrelevant. Our prior deals with pixel information, what was in each patch at each point in time. If there is complex motion, it is represented by the pixels and it is up to the model to understand that motion. Our prior is only there to allow the model to better understand the spatial and temporal dimensions of the data. One case though where our method could potentially be less effective is if the camera moves. In this case yes it might be more difficult for our prior to model motion through time. We’ve mentioned this potential issue in the paper.

---

### Author Response · Authors · 2022-11-16
**Response to reviews**

We thank the reviewers for their comments. The reviewers seem to agree with our motivation that there is a need for good motion priors for video Transformers. They also seem to agree that our paper is overall well written, easy to follow. As for limitations, a remark made by all reviewers is that adding PatchBlender to various types of Transformer architectures and comparing the performance would strengthen the paper. We agree with this statement and are currently working on gathering such results, but we would like to point out a significant bottleneck. Reproducing results in the video domain can be very difficult, specifically when code, pretrained weights and/or hyperparameters are not publicly available in order to reproduce baselines.

We will also add per review comments below.

---

> ### Author Response · Authors · 2022-11-18
> **MViT results added**
>
> We now have MViTv2 results with and without PatchBlender on Something-Something v2, which we've added to the paper. We will keep working on adding further comparative results for MViT and also with other types of Transformer architectures. We've actually been working on this for a while now, but like mentioned previously, reproducing baselines can be time consuming for the mentioned reasons.

---

### Decision · Program_Chairs · 2023-01-20

**Decision:**

Reject

**Justification For Why Not Higher Score:**

 The reviewers raise concerns regarding lack of comparisons of the proposed model with other video adaptations of transformer models, such as TimeSformer, VideoSwin, and UniFormer, or temporal convolutions. They further note that the contributions are marginal.
The rebuttal submitted by the authors does not sufficiently address these concerns. Given the lack of empirical validation of the proposed idea, the paper is not suggested for publication in ICLR.


**Justification For Why Not Lower Score:**

N/A

**Metareview: Summary, Strengths And Weaknesses:**

The paper proposes to learn a generic motion prior, that takes the form of  linear combination of temporal embeddings in video transformers  to help different video-related tasks (motion prediction, video labeling). The approach is validated on motion prediction from a single input frame (synthetic dataset) and video labelling (on two classic benchmarks). The reviewers raise concerns regarding lack of comparisons of the proposed model with other video adaptations of transformer models, such as TimeSformer, VideoSwin, and UniFormer, or temporal convolutions. They further note that the contributions are marginal.
The rebuttal submitted by the authors does not sufficiently address these concerns. Given the lack of empirical validation of the proposed idea, the paper is not suggested for publication in ICLR.


**Summary Of Ac-Reviewer Meeting:**

N/A